# Magnetic Resonance-Guided Reirradiation for Local Recurrence within the Prostate or in the Prostate Bed: One-Year Clinical Results of a Prospective Registry Study

**DOI:** 10.3390/cancers14081943

**Published:** 2022-04-12

**Authors:** Morgan Michalet, Olivier Riou, Jeremy Cottet-Moine, Florence Castan, Sophie Gourgou, Simon Valdenaire, Pierre Debuire, Norbert Ailleres, Roxana Draghici, Marie Charissoux, Carmen Llacer Moscardo, Marie-Pierre Farcy-Jacquet, Pascal Fenoglietto, David Azria

**Affiliations:** 1Montpellier Cancer Institute, University Federation of Radiation Oncology of Mediterranean Occitanie, University Montpellier, INSERM U1194 IRCM, 34298 Montpellier, France; olivier.riou@icm.unicancer.fr (O.R.); jeremy.cottet-moine@icm.unicancer.fr (J.C.-M.); simon.valdenaire@icm.unicancer.fr (S.V.); pierre.debuire@icm.unicancer.fr (P.D.); norbert.ailleres@icm.unicancer.fr (N.A.); roxana.draghici@icm.unicancer.fr (R.D.); marie.charissoux@icm.unicancer.fr (M.C.); carmen.llacer@icm.unicancer.fr (C.L.M.); pascal.fenoglietto@icm.unicancer.fr (P.F.); david.azria@icm.unicancer.fr (D.A.); 2Biometrics Unit ICM, Montpellier Cancer Institute, University Montpellier, 34090 Montpellier, France; florence.castan@icm.unicancer.fr (F.C.); sophie.gourgou@icm.unicancer.fr (S.G.); 3Institut de Cancérologie du Gard, University Federation of Radiation Oncology of Mediterranean Occitanie, CHU Carémeau, 30900 Nîmes, France; mariepierre.farcyjacquet@chu-nimes.fr

**Keywords:** prostate cancer, reirradiation, MRI-guided radiotherapy (MRgRT), stereotactic body radiotherapy (SBRT)

## Abstract

**Simple Summary:**

MRI-guided radiotherapy (MRgRT) is a new technique of radiotherapy. We evaluated this treatment for the reirradiation of patients with local-only recurrence of prostate cancers. The objectives were to evaluate the tolerance and the first clinical results in a cohort of 37 patients with a 1-year follow-up. The treatment was well tolerated with no grade > 2 acute toxicities and only one (3%) grade 3 late toxicity (hematuria). The clinical results were promising with a six, nine and 12-months biochemical-recurrence free survival of 97.3%, 86.5% and 65.0%, respectively. In conclusion, MRI-guided reirradiation might become an interesting option for the treatment of locally recurrent prostate cancers.

**Abstract:**

Around 33% of patients treated by EBRT or brachytherapy will present a biochemical recurrence. SBRT is a new option for the treatment of patients with local-only recurrence. MRgRT seems to be interesting for the treatment of these recurrences. This article presents the one-year late tolerance and biochemical recurrence-free survival results of a prospective registry study. Patients with intraprostatic (or in the prostate bed) recurrence were treated with 5 to 9 fractions (median dose of 30 Gy in 5 fractions) with the MRIdian^®^ system. PSA level and toxicities were evaluated before treatment and at three, six and 12 months after treatment. Thirty-seven patients with a median age of 74.5 years old were treated between 21 October 2019 and 7 December 2020. Acute tolerance was excellent with no grade >2 toxicities. Twelve months after treatment, we observed an increase of grade 1–2 dysuria (46% vs. 13% before treatment) and grade 1 polyuria (73% vs. 7%). The six, nine and 12-months biochemical-recurrence free survival were 97.3%, 86.5% and 65.0%. Fifteen patients (40%) presented a biochemical recurrence. Nine of these 15 patients (60%) had a persistent disease within the treated volume. In conclusion, MRgRT is safe and has promising survival results.

## 1. Introduction 

Overall, 32% of men treated for a localized prostate cancer are treated with external beam radiotherapy (EBRT) or brachytherapy (BT) [1].This corresponds to 23,838 patients in France for the year 2018 [2]. Based on Phoenix criteria (PSA > nadir + 2 ng/mL), we estimate that 20 to 40% of these patients will present a biochemical recurrence [3,4]. With the emergence of new prostate cancer imaging modalities, such as magnetic resonance imaging (MRI) and positron emission tomography/computed tomography (PET/CT) with choline or prostate-specific membrane antigen (PSMA), local-only recurrences are diagnosed in up to 33% of those patients [5]. The historical standard treatment for these recurrences is androgen deprivation therapy (ADT), without possibility of curative intent [6]. A recent meta-analysis suggested good survival results with reirradiation (by stereotactic body radiotherapy (SBRT) or brachytherapy) [7]. Moreover, this treatment seems to generate less genito-urinary (GU) toxicities than other local salvage treatment modalities proposed for this indication (prostatectomy, high-intensity focalized ultrasound, and cryotherapy) [8,9,10]. MRI-guided radiation therapy (MRgRT) is a new promising radiotherapy modality for the treatment of pelvic tumors, especially for prostate cancers [11,12]. Indeed, this technique allows for better tumor volume delineation, thanks to good soft tissue contrast with MRI, as well as a daily adaptation of the dosimetric plan to variations of anatomy with an integrated treatment planning system (TPS), and the possibility of continuous tracking by cine-MRI acquisitions. This technique seems particularly suitable to prostate reirradiation with the need to deliver high doses in a small volume with maximum sparing of pelvic organs at risk (OAR) such as bladder and rectum. Preliminary results were promising in terms of acute tolerance [13]. We present here the one year clinical results in terms of acute and late tolerance and biochemical recurrence-free survival.

## 2. Methods and Materials 

### 2.1. Inclusion Criteria

Inclusion was prospectively proposed to all patients with isolated recurrence within the prostate or in the prostate bed after primary radiotherapy (EBRT or brachytherapy). PET with choline or PSMA was systematically performed to confirm the absence of visible distant metastases or pelvic node involvement. No histology confirmation was required if strong evidence suggested local prostatic cancer recurrence: prostate-specific antigen (PSA) increase confirmed by two consecutive measurements, lesion with high evidence of tumor recurrence on prostate DWI-MRI, and choline or PSMA avidity in prostate. Other inclusion criteria were: Eastern Cooperative Oncology Group (ECOG) performance status = 0 or 1, no previous intestinal or genitourinary radiation-induced toxicity of grade 3 or higher, >12-month interval between the primary EBRT or brachytherapy and adaptive MRgRT, no MRI contraindication (presence of non-MRI compatible implanted cardiac devices, claustrophobia, psychiatric disorders, metal objects), and absence of bilateral hip prostheses (that could alter the treatment plan quality). This study was registered in the French Health Data Hub (registration number: #1802) and was approved by our local research committee (ICM-ART 2020/01). All patients signed an informed consent form before treatment.

### 2.2. Simulation

All patients underwent CT simulation directly followed by MRI simulation using the MRIdian^®^ apparatus (ViewRay Inc. Oakwood Village, OH, United States) to ensure reproducibility of the anatomic configuration. MR and CT images were rigidly registered for target volume delineation, while only the MR images were used for other organs. Furthermore, for dose calculation, CT to MR image registration was performed using an elastic registration algorithm. No contrast agent was needed because soft tissue contrast in MR images was considered sufficient. During the CT simulation, MRI dummy surface coils with similar electron attenuation properties to real MRI coils were placed on the custom immobilization device. MRI images were acquired with a TrueFISP sequence (T1/T2, free breathing, 173 s; resolution of 1.5 mm × 1.5 mm; field of view of 45 cm × 30 cm × 36 cm).

### 2.3. Treatment Planning

The gross tumor volume (GTV) was delineated using the data from the MRI simulation images without injection, MRI diagnostic images, and/or PET with choline (or PSMA) images, when useful. No CTV was defined and an isotropic margin of 3 mm was used from the GTV for the planning target volume (PTV) extension. In all cases, GTV corresponds to a volume smaller than the entire prostate or the prostate bed area. Dose prescription and organs at risk (OAR) dose constraints were determined as described in the GETUG AFU 31 protocol. The initial treatment plan varied from 27.5 Gy in five fractions to 38.7 Gy in nine fractions and was adapted at each session, if necessary. The considered OARs were: rectum (V27 < 2 cc; V12 < 20%), bladder (V27 < 5 cc; V12 < 15%), urethra (V36 < 1 cc, V24 urethra + 3 mm < 30%), and femoral heads. Treatment planning was done using the ViewRay^®^ TPS, with normalization on D50 (100% of the prescribed dose covers 50% of the target volume), while ensuring 95% PTV coverage within the 95% isodose.

### 2.4. Daily Adaptive Treatment Workflow 

Patients were positioned to target the dose to the prostate gland or bed volume using the daily MR images (similar imaging protocol as the one used for simulation). After rigid registration of the GTV, OAR contours were propagated on the daily MR image using deformable image registration. If the OAR contours were not considered optimal, on-line modifications were made by the physician. The initial plan was then evaluated by the physician and the physicist. If all dose constraints were met, no adaptation was required (non-adapted fractions). If a decrease in tumor coverage and/or inacceptable OAR dose constraints were observed, the initial plan was optimized on the integrated TPS (adapted fractions). The electron density map (transferred from the CT to the MR images) and the skin contour were checked to ensure correct dose recalculation. Quality assurance of the newly optimized plan was performed by re-calculating the plan with a secondary Monte Carlo algorithm before irradiation. Tracking was ensured by following a contrasted structure (usually the prostate) on sagittal images obtained by cine MR. The beam was turned off when more than 5% of the tracked structure was outside the threshold of 3 mm from its initial position.

### 2.5. Clinical Assessment, Dosimetric Evaluation and Endpoints

The primary endpoint was the assessment of late toxicities. Secondary endpoints were PSA response, biochemical recurrence-free survival (bRFS), impact of the adaptive treatment on the target volume coverage, and death from any cause. PSA response was defined as the decrease of PSA of more than 30% compared to pre-reirradiation PSA levels. Biochemical progression was defined as an increase of PSA levels of more than 50% compared to post-reirradiation nadir. The bRFS was defined as the time between the end of reirradiation and the biochemical progression. Patients without biochemical recurrence are those with a PSA response or a stable PSA (PSA between minus 30% compared to pre-irradiation PSA level or plus 50% compared to post-therapeutic nadir).

All toxicity events were reported according to the Common Terminology Criteria for Adverse Events (CTCAE) v5.0 at each clinical examination. The International Prostate Symptom Score (IPSS) was filled in at baseline (i.e., at inclusion) and every six months after MRgRT end. According to the IPSS, symptoms were categorized as mild (0–7), moderate (8–19), or severe (20–35). Clinical outcomes and treatment-related toxicity events were assessed and recorded before treatment, on the last day of treatment, and after MRgRT (3, 6, and 12 months). PSA was measured before treatment, and at six weeks, three months, six months, and 12 months after treatment.

For each adapted fraction delivery, the predicted plan (initial plan calculated on the daily image) and the delivered plan (new plan on the daily image) were compared a posteriori with the initial plan. PTV coverage values as well as OAR volumetric doses were recorded.

### 2.6. Statistical Analysis

Quantitative variables were described by the number of observations (N), median, minimum, and maximum, mean and standard deviation. The Wilcoxon signed ranks test was used to compare the distributions of the quantitative variables when they were matched. Categorical variables were described by the number of observations (N) and the frequency (%) of each mode. Missing categories were counted. Percentages were calculated in relation to the total population excluding missing data. The Kaplan-Meier method was used to analyze survival data and to estimate median survival rates and times. The associated survival curves were presented. All statistical tests were two-sided and the significance level was set at 5% (i.e., *p* < 0.05). Statistical analyses were performed with STATA v16.0 and SAS v9.3 software.

## 3. Results

### Initial Patient and Treatment Characteristics

Thirty-seven patients were included from 21 October 2019 to 7 December 2020. The baseline (i.e., at inclusion) patient and tumor characteristics are described in Table 1. The median age was 74.5 years and all patients had a good performance status (ECOG score 0 or 1 in 89.1% and 2 in 10.8%). The median PSA level was 3.36 ng/mL (range, 0.34–34.7) and the median PSA doubling time was 7.20 months (0.80–144.0). The baseline median IPSS score was 6 (0–33).

The primary treatment had been EBRT in 25 patients (67.5%) with or without concomitant ADT, brachytherapy in two patients (5.4%), and EBRT after prostatectomy due to increasing PSA in eight patients (22.2%). The prescribed dose (primary treatment) ranged from 66 to 80 Gy for EBRT (median equivalent dose in 2-Gy fractions, EQD2 = 74 Gy) and 160 Gy for Low Dose Rate (LDR) brachytherapy. The median interval between EBRT/brachytherapy and MRgRT was 88 months (range, 21–240). Two patients (5.4%) had a second treatment before inclusion in this study: EBRT + HIFU for one patient and brachytherapy + EBRT for another one. Eight patients (21.6%) received ADT during MRgRT. Six patients (16.2%) were castration-resistant.

The prescribed dose for MRgRT was 30 Gy in five fractions (*n* = 28, 75.6%), 27.5 Gy in five fractions (*n* = 6, 16.2%), 30 Gy in six fractions (*n* = 2, 5.4%), or 38.7 Gy in nine fractions (*n* = 1, 2.7%). This variation of prescribed dose was at the discretion of the physician and could be the consequence of the prescription of first treatment, the patient’s anatomy, and/or gastro-intestinal or genito-urinary disorders.

## 4. Initial Treatment Plans

The prescription dose was calculated relative to the D50% (50% of the PTV received 100% of the prescription dose) while ensuring PTV V95% ≥ 95% (95% of the PTV received at least 95% of the dose prescription). The median GTV volume was 7.45 cm^3^ (range, 0.88–61.49). All patients met the tumor coverage objectives, with a median PTV V95% of 96% (range, 0.92–0.99). The median PTV V100% was 64% (range, 0.50–1.00) because, for some patients, the normalization had to be adapted to respect the PTV V95% coverage.

All plans met the rectum and bladder dose constraints.

Table 2 presents the median parameters values (for PTV and organs at risk) of the initial plans.

### 4.1. Dosimetric Benefits of Adaptive MRgRT

Twenty-five patients (67.5%) had a daily adaptive treatment. The adaptive treatment significantly improved PTV coverage, median PTV V100% rising from 63% for the predicted plan to 68% for the delivered plan, with a *p*-value of 0.002; and median PTV V95% rising from 94% for the predicted plan to 96% for the delivered plan with a *p*-value < 0.001. This improvement in PTV coverage did not translate into worsened OAR sparing (bladder and rectum). Table 3 presents the comparison of dosimetric parameters between predicted and delivered plans.

### 4.2. Biochemical Response

The median PSA level before reirradiation was 3.38 ng/mL. This level decreased progressively until 6 months after treatment (2.01 ng/mL at six weeks, 1.36 ng/mL at three months and 0.89 ng/mL at six months post-treatment). We observed a slight increase of median PSA levels at 12 months post-treatment (1.05 ng/mL).

Figure 1 presents the bRFS curve after treatment.

The bRFS rates were 97.3%, 86.5% and 65.0% at 3, 6 and 12 months, respectively.

All patients were alive 12 months after treatment.

### 4.3. Acute Toxicities 

At baseline (i.e., at inclusion, before MRgRT), five patients (13%) reported grade 1–2 dysuria, seven patients (19%) reported grade 1–2 urinary incontinence and nine patients (24%) reported grade 1–2 polyuria. No grade > 2 toxicities were reported.

The Table 4 presents the results in terms of acute tolerance for genito-urinary (GU) and gastro-intestinal (GI) toxicities at three time points of evaluation compared to baseline.

MRgRT was well tolerated with no grade > 2 acute toxicity during treatment and the three months following. At the end of MRgRT, there was a small increase of grade 1–2 dysuria (19% vs. 13% before treatment) and grade 1–2 polyuria (38% vs. 24% before treatment).

Six months after treatment, there was an increase of grade 1 polyuria (36% vs. 19% before treatment) without increase of grade 2 or more. One patient experienced a grade 3 hematuria requiring an intervention (hematuria secondary to a tumoral bladder invasion treated with suprapubic catheterization) with good result, a full recovery and no consequences after that.

The median IPSS was stable (6 before treatment and 6.5 at 6 months post-treatment).

## 5. Late Toxicities 

Twelve months after treatment, we observed an increase of grade 1–2 dysuria (12 patients (32%) reported a grade 1 dysuria and 5 patients (14%) a grade 2) and grade 1 polyuria (27 patients (73%)).

There were no GI toxicities and no grade > 2 GU toxicities.

The median IPSS was stable (6 before treatment and 5.5 at 12 months post-treatment).

Table 5 presents the results of late tolerance 12 months after treatment.

### Topography of Recurrences

Fifteen patients (40%) presented a biochemical recurrence during follow-up. All patients presenting a recurrence had a choline-PET/CT +/− PSMA-PET/CT for the diagnostic of the topography of the recurrence. Six of these patients (40%) presented an intraprostatic persistent or progressive disease within the treated volume. Figure 2 shows an example of choline-PET/CT performed nine months after treatment with persistent disease, registered with the MRIdian^®^ simulation image showing the treated PTV during the reirradiation.

Two patients (14%) presented an isolated biochemical recurrence. One patient (7%) presented a recurrence inside the prostate but remotely from the reirradiation volume. One patient (7%) presented a recurrence inside the right seminal vesicle. Two patients (13%) presented a pelvic lymph node recurrence. Two patients (14%) presented a bone metastatic evolution with persistent intraprostatic disease. One patient (7%) presented a pelvic lymph node evolution with persistent intraprostatic disease. Finally, nine patients (60%) had a persistent disease within the treated volume.

## 6. Discussion

This study showed promising results for the treatment of the local relapse of prostate cancer with MRgRT. The one-year biochemical recurrence-free survival (BRFS) was 86.5% at six months and 65.0% at 12 months. The treatment was well tolerated with only one patient experiencing a grade 3 acute toxicity (hematuria) and no grade > 2 late genitourinary and gastrointestinal toxicities.

This is, to our knowledge, the first study presenting survival and late tolerance results of this technique. Some studies evaluated the results of SBRT as a salvage treatment for intraprostatic relapses, with prescriptions varying between 100 and 145 Gy with LDR brachytherapy, 19 and 36 Gy with HDR brachytherapy in one to five fractions and 25 to 36 Gy with hypofractionated EBRT in five to six fractions [14].

We confirmed our previous results demonstrating the dosimetric feasibility and good acute tolerance of this treatment. In the present study on 37 patients, 25 had an adaptive treatment. All patients met the initial dosimetric objectives (PTV coverage and OAR dose constraints). Adaptation led to a significantly better PTV V95% and PTV V100% coverage (from 94% to 96% and 63% to 68% respectively), with respect to OAR dose constraints, as previously described [13]. We also confirmed the good acute tolerance with 39% of grade 1–2 polyuria and 20% of grade 1–2 dysuria at six months. Only one patient experienced a grade 3 toxicity (hematuria secondary to a tumoral bladder invasion which was resolved after bladder lavage by suprapubic catheterization). There was no gastrointestinal (GI) toxicities and no worsening of the urinary incontinence rate. This is consistent with other studies of EBRT for salvage reirradiation with no or few grade 3 toxicities and around 20% of grade 1–2 genitourinary (GU) toxicities [15,16,17].

The recent publication of the MASTER meta-analysis showed that SBRT and brachytherapy (HDR or LDR) are probably the best treatments for intraprostatic relapse of prostate cancers first treated with EBRT or brachytherapy. The two-year recurrence-free survival (RFS) of SBRT and brachytherapy were similar to radical prostatectomy (RP) and cryotherapy (from 58% to 79%, non-significant difference) but SBRT and brachytherapy induced less severe GU toxicities than RP, cryotherapy and HIFU (*p* < 0.002) [7]. In our study, the three, six and 12-months bRFS were 97.3%, 86.5% and 65.0%, respectively, which is consistent with published data. In a recent systematic review, the 2-year bRFS was 71%, 74% and 54.9% for the LDR-brachytherapy studies, HDR-brachytherapy studies and SBRT studies, respectively [18]. Focusing with other SBRT studies, the 2-year bRFS varied between 41.7% and 80% and the 1-year bRFS was around 80% [15,16,17,19,20,21,22,23,24,25,26]. Moreover, in our study, we decided to treat and include patients regardless of the PSA doubling time. Some authors identified a cutoff of 10–12 months as a bad prognostic factor, with more biochemical recurrences [27,28,29]. In our study, the median PSA doubling time of 7.2 months could also explain a lower bRFS.

In our patient population, many patients (10/37) had a pre-therapeutic PSA > 2 ng/mL (patients with recurrences within the prostate bed or patients who had a choline-PET/CT for a rising PSA < nadir + 2 ng/mL or patients with ADT). Phoenix criteria were therefore not useable. Moreover, we think that in the situation of salvage treatment, these criteria are not suitable. We decided to create a new PSA response outcome to define bRFS results after reirradiation. Progression was defined by the increase of PSA level of more than 50% compared to post-reirradiation nadir level and response as a decrease of PSA level of more than 30% compared to the pre-reirradiation level. This difference could make the comparison between our study and other studies incorrect and explain our lower bRFS. Based on this definition, we reported a one-year bRFS of 65%.

As compared to other studies of prostate SBRT reirradiation, we had differences in patient selection, as 21.6% of our patients previously had RP and a prostate bed irradiation before salvage reirradiation. Moreover, only 21% of our patients had concomitant androgen deprivation therapy (ADT), while this rate was as high as 48% in other studies [23,24].

Differences in treatment planning might make our results difficult to compare to other prostate SBRT reirradiation studies. Indeed, we decided to treat the prostate partially, only on the GTV defined by choline-PET/CT or PSMA-PET/CT and MRI, contrary to other authors who chose to retreat the whole prostate [17,19,20,24]. Given that this salvage treatment is not a standard treatment, with possible severe long term toxicities, we chose to only treat the recurrence site, as in the GETUF AFU 31 trial [30]. We also chose to prescribe 30 Gy in five fractions of 6 Gy when some other authors treated patients with six fractions of 6 Gy [25,26]. At the time of the study, the GETUG AFU 31 trial (phase I/II) was the only protocol considered as the “standard” in France by the ethical committees. Therefore, we decided to start with ~30 Gy in five fractions because this schedule was considered “acceptable” based on the interim analysis results. We could not include our patients in the GETUG-AFU 31 trial because MRIdian Linac^®^-based treatments were not allowed. Therefore, we decided to wait to have a longer follow-up for our first patients before performing a dose escalation. From now on, based on our favorable tolerance results and following the update of the GETUG-AFU 31 trial, we decided to treat these patients with a regimen of 6 fractions of 6 Gy.

We evaluated the late tolerance with an assessment of 1-year GU and GI toxicities. The treatment was well tolerated with no grade > 2 toxicities. This was consistent with other salvage SBRT data in literature where grade > 2 GU toxicities varied between 1.0% and 8.0%. In contrast, we did not observe any GI toxicities when other SBRT studies reported grade 1–2 GI toxicities from 1.0% to 31.6% [15,17,20,21,22,23,25]. The lower toxicity rate in our study could be explained by reduced volume and lower total dose compared with protocols where the entire prostate receive higher doses. These results are also probably related to the benefits of MRgRT thanks to the continuous tracking with cine-MR acquisition and daily adaptation, allowing for reduced treatment margins and lower rectal doses [12,31]. Brachytherapy (LDR or HDR) seems to lead to higher rates of late severe GU toxicities (grade > 2 from 9.0% to 30.0%) [27,32,33,34,35,36].

We decided to be permissive on urethra dose constraints (usually V36 < 1cc and V24 urethra + 3 mm < 30%) as this would have led to decrease some PTV coverage for lesions close to the urethra, and no urethra-related toxicity was previously described in the feasibility study of MRgRT [13]. We will carefully follow the later tolerance results and readjust our habits if late urethra-related toxicities are observed.

A careful selection of patients is required for the treatment of local prostate relapses. An ESTRO ACROP Delphi consensus was recently published concerning initial evaluation, diagnostic tests and salvage treatment [37]. Consensus items obtained were: no maximum age; ECOG score 0 or 1; IPSS should be known, maximum Qmax at salvage: no minimum inferior value; CTV: GTV defined on mpMRI plus adaptive margin; imaging for metastatic disease: choline-PET required; ADT should not be delivered concomitantly with re-irradiation; Phoenix definition of biochemical relapse is valid for re-treated patients; and primary treatment dose should always be considered when deciding salvage SBRT dose. All of these items are consistent with our patient selection criteria and patient evaluation, except for the definition of biochemical relapse. Another future suitable outcome for those studies could be the time without ADT.

Our study presents some limits. First, this is a monocentric study. Indeed, MRgRT is a new technique and at the time of the study, only a few machines were available for the treatment of salvage local prostate relapse. Second, our sample size is limited but is consistent with other prostate SBRT reirradiation studies. Longer inclusion times and follow-up are warranted to confirm our favorable results, especially for late GU tolerance.

## 7. Conclusions

MRgRT is a good option for the salvage treatment of intraprostatic or within the prostate bed relapse. Clinical results are encouraging with good one-year bRFS. The treatment is well tolerated with favorable acute and one-year tolerance. A longer follow-up is needed to assess late toxicities as well as to define prognostic and predictive markers for a better patient selection.

## Figures and Tables

**Figure 1 cancers-14-01943-f001:**
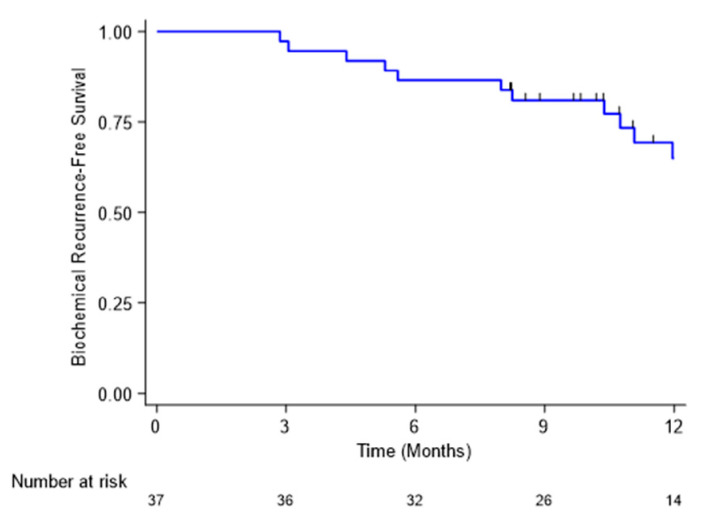
Biochemical recurrence-free survival curve. Recurrence was defined as an increase of PSA level of more than 50% compared to nadir post-reirradiation.

**Figure 2 cancers-14-01943-f002:**
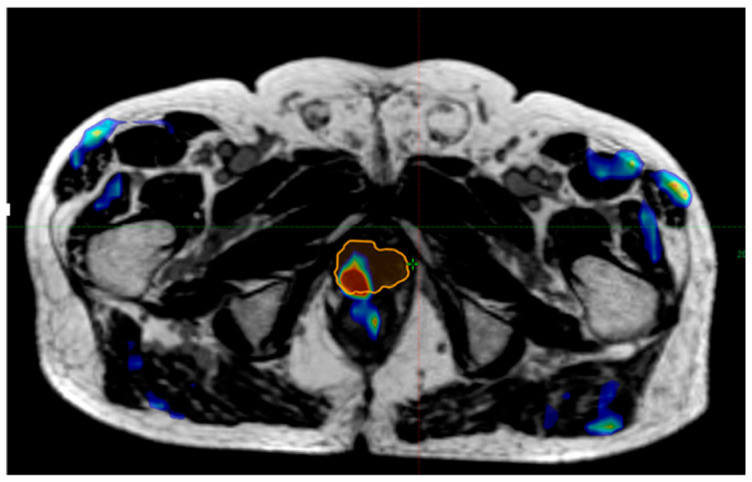
Example of an intraprostatic persistent disease. This patient presented a PSA level increase of more than 50% compared to post-reirradiation nadir at 9 months post-treatment. A registration was done between the MRIdian^®^ simulation image for reirradiation with PTV in orange line and the Choline-PET/CT performed after this rising PSA. This registration highlighted a persistent hyperfixation within the treated volume.

**Table 1 cancers-14-01943-t001:** Patient and treatment baseline characteristics.

Characteristics	*N* = 37	%
Age		
Median (range, in years)	74.5 (56–93)	
**ISUP group before the primary treatment**		
1	10	27.8
2	7	19.4
3	10	27.8
4	6	16.7
5	3	8.3
Unknown	1	
**Primary treatment techniques**		
EBRT or EBRT + ADT	25	67.5
Brachytherapy	2	5.4
Prostatectomy + EBRT	8	22.2
Other (EBRT + HIFU or BT + EBRT)	2	5.4
**Irradiation dose delivered during the primary treatment**		
Median (range, in Gy)	74.0 (62.0–180)	
Median time between the primary treatment and the re-irradiation (range, in months)	88.0 (21–240)	
**WHO score before re-irradiation**		
0	15	40.5
1	18	48.6
2	4	10.8
**PSA level (ng/mL) before re-irradiation**		
Median (range)	3.36 (0.34–34.7)	
PSA doubling time (range, in months)	7.20 (0.80–144.0)	
**IPSS score and symptom groups before re-irradiation**		
Median score (range)	6 (0–33)	
Mild (1–7)	13	35.1
Moderate (8–19)	5	13.5
Severe (20–35)	3	8.1
Unknown	16	43.2
**Dose prescription**		
27.5 Gy/5 fractions	6	16.2
30 Gy/5 fractions	28	75.6
30 Gy/6 fractions	2	5.4
38.7 Gy/9 fractions	1	2.7
**ADT sensitivity**		
Hormone sensitive	31	83.8
Castration resistant	6	16.2
**Concomitant ADT during re-irradiation**		
Yes	8	21
No	30	79
**Adaptive treatment**		
Yes	25	67.6
No	12	32.4
**Treatment duration by fraction**		
Median (range, in minutes)	42 (30–95)	
**Total treatment duration**		
Median (in days)	11 (9–31)	

Abbreviation: ADT = androgen deprivation therapy; BT = brachytherapy; EBRT = external beam radiation therapy; HIFU = high intensity focal ultrasound; IPSS = international prostate score symptom; ISUP = international society of urological pathology; PSA = prostate specific antigen; WHO = the World Health Organization.

**Table 2 cancers-14-01943-t002:** Planned dose parameters for MR guided re-irradiation.

Parameters	Median (Range)
PTV V100% (%)	64 (50–100)
PTV V95% (%)	96 (92–99)
Rectum V27 (cm^3^)	0.20 (0.00–1.83)
Rectum V12 (%)	13 (0–20)
Bladder V27 (cm^3^)	0.01 (0.00–4.97)
Bladder V12 (%)	3 (0–15)

**Table 3 cancers-14-01943-t003:** Comparison of dosimetric parameters between predicted and delivered plans for patients with plan adaptation.

Parameters	Predicted Plans: Median (Range)	Delivered Plans: Median (Range)	Difference (*p*-Value)
PTV V100% (%)	63 (31–1.00)	68 (54–100)	0.002
PTV V95% (%)	94 (89-97)	96 (93–98)	< 0.001
Rectum V27 (cm^3^)	0.51 (0.00–1.43)	0.46 (0.00–1.42)	0.893
Rectum V12 (%)	14 (0–21)	14 (0–19)	0.403
Bladder V27 (cm^3^)	0.17 (0.00–3.58)	0.26 (0.00–3.84)	0.468
Bladder V12 (%)	5 (0–17)	5 (0–14)	0.614

**Table 4 cancers-14-01943-t004:** Acute GU and GI toxicities according to CTCAE v5.0.

Toxicity	Before MRgRT (Number of Patients)	Last Day of MRgRT (Number of Patients)	Three Months After MRgRT (Number of Patients)	Six Months After MRgRT (Number of Patients)
**Dysuria**				
**g0**	32 (86%)	30 (81%)	25 (83%)	29 (80%)
**g1**	3 (8%)	3 (8%)	5 (17%)	6 (17%)
**g2**	2 (5%)	4 (11%)	0 (0%)	1 (3%)
**Missing data**	0	0	7	7
**Hematuria**				
**g0**	35 (95%)	34 (92%)	28 (94%)	33 (91%)
**g1**	2 (5%)	2 (5%)	1 (3%)	1 (3%)
**g2**	0 (%)	1 (3%)	1 (3%)	1 (3%)
**g3**	0 (%)	0 (%)	0 (%)	1 (3%)
**Missing data**	0	0	7	1
**Urinary incontinence**				
**g0**	30 (81%)	30 (81%)	27 (90%)	30 (83%)
**g1**	5 (14%)	4 (11%)	3 (10%)	5 (14%)
**g2**	2 (5%)	3 (8%)	0 (0%)	1 (3%)
**Missing data**	0	0	7	1
**Polyuria**				
**g0**	28 (76%)	23 (62%)	24 (80%)	22 (61%)
**g1**	7 (19%)	9 (24%)	5 (17%)	13 (36%)
**g2**	2 (5%)	5 (14%)	1 (3%)	1 (3%)
**Missing data**	0	0	7	1
**Urinary pain**				
**g0**	37 (100%)	35 (95%)	30 (100%)	34 (94%)
**g1**	0 (0%)	2 (5%)	0 (0%)	1 (3%)
**g2**	0 (0%)	0 (0%)	0 (0%)	1 (3%)
**Missing data**	0	0	7	1
**Diarrhea**				
**g0**	35 (95%)	33 (89%)	30 (100%)	18 (90%)
**g1**	2 (5%)	3 (8%)	0 (0%)	2 (10%)
**g2**	0 (0%)	1 (0%)	0 (0%)	0 (0%)
**Missing data**	0	0	7	1
**Rectal bleeding**				
**g0**	35 (95%)	37 (100%)	30 (100%)	36 (100%)
**g1**	2 (5%)	0 (0%)	0 (0%)	0 (0%)
**g2**	0 (0%)	0 (0%)	0 (0%)	0 (0%)
**Missing data**	0	0	7	1
**Rectal pain**				
**g0**	37 (100%)		30 (100%)	36 (100%)
**g1**	0 (0%)	37 (100%)	0 (0%)	0 (0%)
**g2**	0 (0%)	0 (0%)	0 (0%)	0 (0%)
**Missing data**	0	0 (0%)	7	1

Abbreviations: GU = genitourinary; GI = gastrointestinal; CTCAE = Common terminology criteria for adverse events; g = grade; MRgRT = Magnetic resonance-guided radiotherapy.

**Table 5 cancers-14-01943-t005:** Late GU and GI toxicities according to CTCAE v5.0.

Toxicity	Before MRgRT (Number of Patients)	12 Months After MRgRT (Number of Patients)
**Dysuria**		
**g0**	32 (86%)	20 (54%)
**g1**	3 (8%)	12 (32%)
**g2**	2 (5%)	5 (14%)
**Hematuria**		
**g0**	35 (95%)	36 (97%)
**g1**	2 (5%)	1 (3%)
**g2**	0 (%)	0 (0%)
**Urinary incontinence**		
**g0**	30 (81%)	31 (84%)
**g1**	5 (14%)	4 (11%)
**g2**	2 (5%)	2 (5%)
**Polyuria**		
**g0**	28 (76%)	9 (24%)
**g1**	7 (19%)	27 (73%)
**g2**	2 (5%)	1 (3%)
**Urinary pain**		
**g0**	37 (100%)	35 (94%)
**g1**	0 (0%)	1 (3%)
**g2**	0 (0%)	1 (3%)
**Diarrhea**		
**g0**	35 (95%)	37 (100%)
**g1**	2 (5%)	0 (0%)
**g2**	0 (0%)	0 (0%)
**Rectal bleeding**		
**g0**	35 (95%)	37 (100%)
**g1**	2 (5%)	0 (0%)
**g2**	0 (0%)	0 (0%)
**Rectal pain**		
**g0**	37 (100%)	37 (100%)
**g1**	0 (0%)	0 (0%)
**g2**	0 (0%)	0 (0%)

Abbreviations: GU = genitourinary; GI = gastrointestinal; CTCAE = Common terminology criteria for adverse events; g = grade; MRgRT = Magnetic resonance-guided radiotherapy.

## Data Availability

The data presented in this study are available in this article.

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
