# Peer review of "Magnetic Resonance-Guided Reirradiation for Local Recurrence within the Prostate or in the Prostate Bed: One-Year Clinical Results of a Prospective Registry Study"

_cancers, 2022, doi:10.3390/cancers14081943_

Round 1
Reviewer 1 Report
Thanks to the authors for this beautiful work,
A statistical question about the results, did you make a linear model or multivariable linear mix-effects model to describe the change of IPSS, including prostate volume, age or do you think it is not necessary ?
Reviewer 2 Report
This paper demonstrated the novel clinical course of patients with intra-prostatic/prostate bed recurrence after frontline therapy, by using salvage MRI-guided radiation therapy (MRgRT), the MRIdian® system, which has recently emerged with great attention.
Proposed methodology is a quite innovate model thus far and few studies saw long-enough data regarding bRFS or OS and acute/late toxicities.
As authors suggested, this approach has an advantage that the promising local control is expected in terms of highly conformal dose prescription while precise OAR protection.
Additionally, they developed unique definition of PSA progression following MRgRT reirradiation by the increase of PSA level of more than 50% compared to post-reirradiation nadir level, not favoring their own results compared to other reports.
Fundamentally I agree with their assertion that introduced strategy will contribute to facilitate future coping with focal lesion detected by next-generation imaging tools such as choline-PET/CT or PSMA-PET/CT.
One of the intrinsic weak points of this study includes small sample size in single French institution as feasible clinical data is limited to establish satisfactory evidence. However, the nature of extremely limited system-produced clinical outcome makes analysis using larger patient data difficult at this time, therefore, forthcoming investigation to more patient populations in other areas will be essential to wider validation.
Considering the study carried out according to an appropriate process, this study has potential to be published with a modest amount of revision as per my comments.
I have the following comments:
- Some explain about CTV may be helpful for readers in Treatment planning.
- The authors do not mention initial and/or pretreatment Gleason grade at all. Since Gleason sum definitely affects local control, it brings very useful information for understanding the validity of this method. If authors have the information at hand, they may want to refer to it.
- Present study population consists of very heterogenous patient subgroups. At least, since intraprostatic recurrence after definitive prostate irradiation and intra-prostate bed (moreover following biochemical recurrence) after total prostatectomy are essentially different pathologies, the bRFS should be considered separately. Kaplan-Meier curves must be drawn separately in addition to the all-together one described in the text.
- In Results, in Initial patient and treatment characteristics, prescribed dose fractionation varied substantially. The reasons for choosing these different fractions must be made clear.
- The median PSADT described here was less than 10, yet it would be interesting to discuss the relationship with this number, which is generally seen as a poor prognostic factor leading to distant recurrence.
- There are errors of abbreviation, grammatical mistake, and less orderly language in places.
